

# The impact of tropospheric blockings on duration of the sudden stratospheric warmings in boreal winter 2023/24

Ekaterina Vorobeva[1] and Yvan Orsolini[1]

[1]NILU - The Climate and Environmental Research Institute, Kjeller, Norway

**Correspondence:** Ekaterina Vorobeva (evor@nilu.no)

**Abstract.**

The winter 2023/24 exhibited remarkable stratospheric dynamics with multiple sudden stratospheric warmings (SSWs). Based on the fifth generation European Centre for Medium-Range Weather Forecasts (ECMWF) reanalysis (ERA5) polar-cap averaged 10 hPa zonal wind, three major SSWs are identified - an extremely rare occurrence in a single winter. Two of three SSWs were short-lived, lasting under 7 days. In this study, we give an overview of the three SSWs that occurred in winter 2023/24 and focus on the impact of tropospheric forcing on their duration. Blocking high-pressure systems are shown to modulate wave activity flux into the stratosphere through interactions with tropospheric planetary waves, depending on their location. The rapid termination of the first SSW (14–19 January 2024) is linked to a developing high-pressure system over the North Pacific. The second SSW (16–22 February 2024) terminated quickly due to more contributing factors, one of which was a high-pressure system developed over the Far East. The third SSW (3–28 March 2024) was a long-duration canonical event extending to levels below 100 hPa. In contrast to two short-lived SSWs in winter 2023/24, the tropospheric forcing was sustained around the SSW onset in March 2024, allowing a long event to develop. We also note that conditions for these SSWs were particularly favorable due to external factors, including an Easterly Quasi-Biennial Oscillation (QBO), the presence of El Niño conditions of the ENSO cycle, and the proximity to the solar maximum.

## 1 Introduction

The wintertime high-latitude stratospheric circulation is characterized by prominent eastward winds encircling the pole and spanning from $\sim 100$ to over 1 hPa, also known as the stratospheric polar vortex. The prevailing wintertime stratospheric wind pattern is regularly disturbed by planetary waves (PWs) originating from the troposphere (Matsuno, 1970). Extreme examples of the stratospheric polar vortex disturbance by PWs are sudden stratospheric warmings (SSWs) (see, e.g., Waugh and Polvani, 2010; Charlton and Polvani, 2007). According to the previous studies (see, e.g., Limpasuvan et al., 2016; Shepherd et al., 2014, and references therein), SSWs originate in the upper mesosphere at high latitudes and extend downwards into the stratosphere at high latitudes, before extending further into lower latitudes. SSWs can be classified into minor and major events based on





either a temporary weakening or a reversal of stratospheric winds to a summer-like westward regime, as well as into split and displacement events based on the spatial structure of the disturbed stratospheric polar vortex. Major SSW events tend to occur around every two years on average, however, their occurrence frequency can vary significantly on inter-annual to decadal time scales (Butler et al., 2017).

Most definitions of the SSW onset are based on the stratospheric zonal-mean zonal wind and temperature at 10 hPa (see, e.g., Butler et al., 2015; Baldwin et al., 2021, for reviews). However, it is important to note that these dramatic disruptions of atmospheric circulation impact the entire atmospheric column, spanning from the troposphere to the thermosphere across diverse latitudinal ranges (see, e.g., Limpasuvan et al., 2016; Pedatella, 2023). Duration wise, the eastward wind reversal (typically at 10 hPa) during major SSWs displays remarkable diversity, ranging from short events (lasting only a few days) to more canonical long-lasting events (lasting over several weeks). Previous studies have demonstrated that long SSW events may affect tropospheric weather and climate patterns. The impacts include the occurrence of cold air outbreaks across North America and Eurasia (Kolstad et al., 2010), shifts in the jet stream southward over the Euro-Atlantic region, and a negative phase of the North Atlantic Oscillation (Butler et al., 2017).

Recent studies have emphasized the critical role of tropospheric blocking in the stratospheric variability. Blocking events, characterized by persistent high-pressure systems (hereafter blocking highs or BHs) that disrupt the typical eastward flow, can act as a precursor to SSWs by influencing PWs propagation into the stratosphere (e.g. Martius et al., 2009; Bancalá et al., 2012; Nishii et al., 2011). The geographical positioning of BHs prior to the SSW onset has been studied in the context of SSW types, intensity and duration. Martius et al. (2009) and Bancalá et al. (2012) found a strong relation between the SSW spatial type and the geographical location of BHs in the troposphere, with displacement events being associated predominantly with BHs in the Euro-Atlantic sector (where a climatological PW ridge is located), and split events linked to BHs either in the Pacific sector (climatological PW trough) or simultaneously across both the Atlantic and Pacific sectors. Nishii et al. (2011) demonstrated that BHs over the Euro-Atlantic region tend to increase upward propagation of planetary waves and cause the weakening of the polar vortex, whereas BHs over the western Pacific and Far East suppress the upward propagation of planetary waves and lead to a stronger polar vortex. Orsolini et al. (2018) showed that the synoptic evolution of BHs around onset is a critical parameter determining the SSW duration. BHs situated over the Euro-Atlantic sector are often associated with prolonged SSWs by sustaining the upward wave activity flux, whereas those developing over the western Pacific sector may contribute to the short duration and termination of SSWs by lowering the wave activity flux.

Despite extensive prior research on the predictability of SSWs, it remains an open question why some SSWs develop into long events while others terminate quickly. Addressing this question is the main focus and novel aspect of this paper: the winter 2023/24 presented a beneficial scenario due to the occurrence of multiple SSWs of different durations within 3 months. In winter 2023/24, the stratospheric polar vortex was weak and highly variable, and three major SSWs took place. Two short- and one long-lived events occurred on 14 January, 16 February, and 3 March 2024, respectively. This number of SSWs within one winter is uncommon and could be considered as an extreme occurrence (Ineson et al., 2024). In this study, we give an overview of these SSWs and focus on the impact of the tropospheric forcing on their duration using the fifth generation European Centre for Medium-Range Weather Forecasts (ECMWF) reanalysis (ERA5). For all three SSW events, it is demonstrated that blocking





highs developing over the PW ridges were indeed responsible for the enhanced PWs activity prior to the onset. Short duration

of SSWs in January and February 2024 is shown to be the result of the blocking highs developing in the PWs trough over

the western Pacific / Far East that suppressed the PWs activity and caused a quick termination of these events. In contrast, the

persistent blocking highs over the Euro-Atlantic sector sustained the upward wave activity flux allowing a long SSW in March

2024.

Data and methods used in this study are introduced in Sect. 2. Analysis of multiple SSWs in winter 2023/24 as well as an

65 insight into their tropospheric forcing is presented in Sect. 3. The results obtained are further discussed in Sect. 4.

## 2    Data and Methods

In this study, atmospheric data are obtained from the ECMWF ERA5 reanalysis with 1 h temporal and $0.25° \times 0.25°$ spatial

resolution. Open access ERA5 data are available on 37 pressure levels up to 1 hPa. For an overview of the ERA5 strengths,

biases, and validation, we refer to Hersbach et al. (2020) and Bell et al. (2021).

To assess the stratospheric state and its interaction with the troposphere, the following atmospheric data are used: air tem-

perature (T), zonal and meridional winds (U and V), and geopotential height (z) at all available pressure levels from 1000 to 1

hPa. The aforementioned variables are extracted from the open access ECMWF archive (Hersbach et al., 2023) and averaged

daily. Daily climatological means for T, U, V and z are calculated based on 2000–2022 ERA5 data. For the leap years, 29

February is not considered.

To analyze tropospheric forcing, we calculate the meridional eddy heat flux, $v'T'$, that is directly related to the upward

wave activity flux from the troposphere into the stratosphere (Newman and Nash, 2000). Here, prime denotes the deviation

from the zonal mean. When zonally averaged, the eddy heat flux is proportional to the vertical component of the Eliassen-

Palm (E-P) flux (Andrews et al., 1987). Previous studies have shown that increased (reduced) meridional eddy heat flux may

weaken (strengthen) the stratospheric polar vortex (see e.g. Coy et al., 1997; Newman et al., 2001; Matthias et al., 2016). It

is, therefore, often used as a precursor for weak and strong polar vortex events (Polvani and Waugh, 2004). Typically, $v'T'$ is

calculated at 100 hPa. Hinssen and Ambaum (2010) demonstrated that nearly half of the year-to-year variations in the Northern

Hemisphere stratosphere are influenced by fluctuations in the heat flux at 100 hPa. Contribution to the meridional eddy heat

flux by individual waves is calculated by multiplying the corresponding Fourier components of $v'$ and $T'$ (Newman and Nash,

2000). Thus, zonal wavenumber 1 component of the $v'T'$ is calculated as $v'T'_{wv=1} = v'_{wv=1}T'_{wv=1}$. In this study, we consider

the meridional eddy heat flux area-averaged poleward of $45°N$ (hereafter $[v'T']$).

Various criteria exist to determine the onset of a SSW (Butler et al., 2015). The commonly used definition is based on the

reversal of the zonal-mean U at $60°N$ and 10 hPa as proposed by Charlton and Polvani (2007). However, Butler and Gerber

(2018) focused on optimizing the SSW definition and concluded that features of major SSWs are maximized between 55 and

70 N in the middle stratosphere (30–5 hPa). Therefore, one can miss short and weak major SSW events that do not have

90 enough time to extend their features to $60°N$ when using the classical definition (an example will be shown below in mid-

February 2024). For this reason, the SSW onsets in this study are obtained based on the time series of 10 hPa U area-averaged



poleward of 60°N (hereafter polar-cap averaged) allowing one to assess the polar stratospheric dynamics without a focus on one specific latitude. The SSW onset is defined as the day when daily-mean polar-cap averaged U turns westward, and its duration corresponds to the number of days it remains westward.

## 3 Results

### 3.1 Onset and duration

As mentioned in Sect. 1, the SSW signatures emerge primarily at the high latitudes and extend toward the middle latitudes. Therefore, we focus on analyzing the latitude-time cross-section of the zonal-mean T and U at 10 hPa during December 2023 – April 2024 (Fig. 1). Figure 1 also shows time series of the zonal-mean 10 hPa T and U at 60°N, 65°N and polar-cap averaged.

At the end of December 2023, the polar vortex exhibited signs of weakening across a broad band of latitudes, nearly reversing its flow at the pole (Fig. 1c). Due to the lack of a westward reversal, this weakening does not qualify as a SSW, although it played a key role in the unusually early generation of the two-day wave in the austral hemisphere (Qin et al., 2025). During the first week of January 2024, the temperature exceeded 240 K near the pole, however the westward zonal-mean U did not reach mid-latitudes. Soon after, the zonal-mean U reversed to a westward regime around the pole once again, and spread towards 60°N within the following days. At 60°N, the zonal-mean U remained westward for 2 days only (blue curve in Fig. 1d), while the polar-cap averaged U remained westward until 20 January 2024. Recovery of the polar vortex began thereafter, with the vortex returning to its climatological mean by 24 January 2024 and intensifying in the following two weeks. We identify the SSW in January 2024 as a major event with onset on 14 January 2024, that lasted for 6 consecutive days.

On 15 February 2024, the zonal-mean U reversed at the pole, and the westward U spread towards mid-latitudes within the following days (Fig. 1c). In this case, using the standard SSW definition would identify February SSW as a minor SSW as it didn't have enough time to extend its signatures to 60°N (Fig. 1d). However, we argue that SSW in February 2024 was clearly associated with an increased temperature exceeding 230 K and a reversal of zonal-mean U in a broad range of latitudes over a course of around one week (Fig. 1a,c). By the end of February 2024, the polar vortex nearly came back to its climatological mean. We identify the SSW in February 2024 as a major event with onset on 16 February 2024, that lasted for 7 consecutive days.

Figures 1a,c show a nearly simultaneous reversal of the zonal-mean U in 50–90 N band and the associated increase in the zonal-mean T values over 230 K on 3 March 2024. Compared to the SSWs in January and February 2024, the SSW in March 2024 was a long-lasting event with polar-cap averaged U reaching below -20 m/s, close to the absolute minimum in the 2000–2022 period (Fig. 1d). We conclude that the SSW in March 2024 was a major event with onset on 3 March 2024, that lasted for 26 consecutive days. This event is considered major rather than final warming as U returned to its eastward regime well prior the end of April.





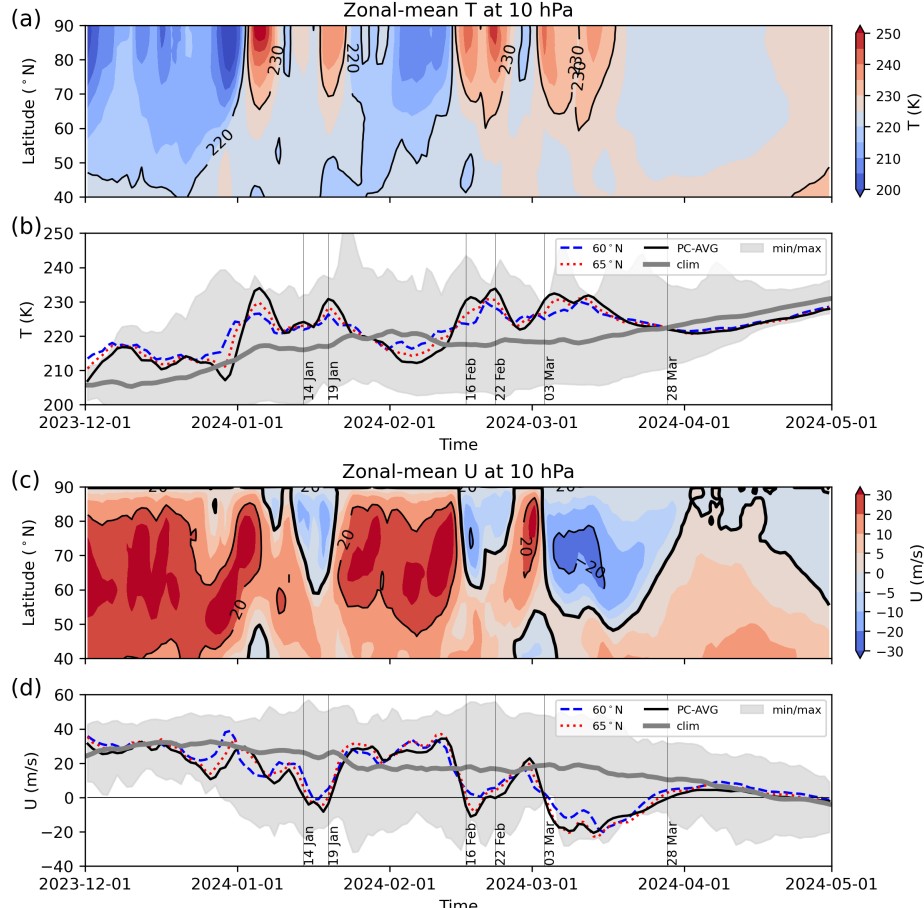

**Figure 1.** Latitude-time cross-section of zonal-mean a) temperature, T (K), and c) zonal wind, U (m/s), at 10 hPa during December 2023 – April 2024. Thick black contours in panel c) indicate zero wind. The associated time series of the zonal-mean b) temperature and d) zonal wind at 60°N (blue dashed), 65°N (red dotted), and area-averaged poleward of 60°N (PC-AVG, black solid). Climatological means for the polar-cap averaged T and U are plotted as thick gray curves and min/max range is shaded based on 2000 – 2022 ERA5 data.

### 3.2 Time-pressure cross-section

This section focuses on the altitude-time evolution of the polar-cap averaged U and meridional eddy heat flux in winter 2023/24. Figure 2a shows that the polar-cap averaged U reversed to the summer-like westward regime three times in the upper strato-
125 sphere in January 2024. First, the 1 hPa polar-cap averaged U reversed on 2 January 2024 and caused weakening of 10 hPa polar-cap averaged U around 5 January 2024 as was also seen in Fig. 1. At 10 hPa, polar-cap averaged U reversed on 14 January 2024 and remained westward until 20 January 2024, while one can see two separate episodes of 1 hPa polar-cap averaged U reversal on 11–14 January and 17–21 January 2024 that reached down to around 20 hPa. In February 2024, the polar-cap averaged U reversal occurred nearly simultaneously in a deep layer of 20–1 hPa. By 24 February 2024, polar-cap averaged U



returned to the eastward regime. In March 2024, polar-cap averaged U reversed in the entire stratosphere (100–1 hPa). From
Fig. 2a, one can see that the westward polar-cap averaged U lasted longer at 10 hPa pressure level (up to 29 March 2024) while
1 hPa polar-cap averaged U returned to eastward regime already on 16 March 2024. One can see two episodes of the westward
polar-cap averaged U intensification located at slightly different pressure levels that are separated in time by approximately a
week. Another interesting feature seen here is that the short-lived SSW in January 2024 was more intense than the long SSW

in March 2024 in terms of westward polar-cap averaged U magnitude at 1 hPa.

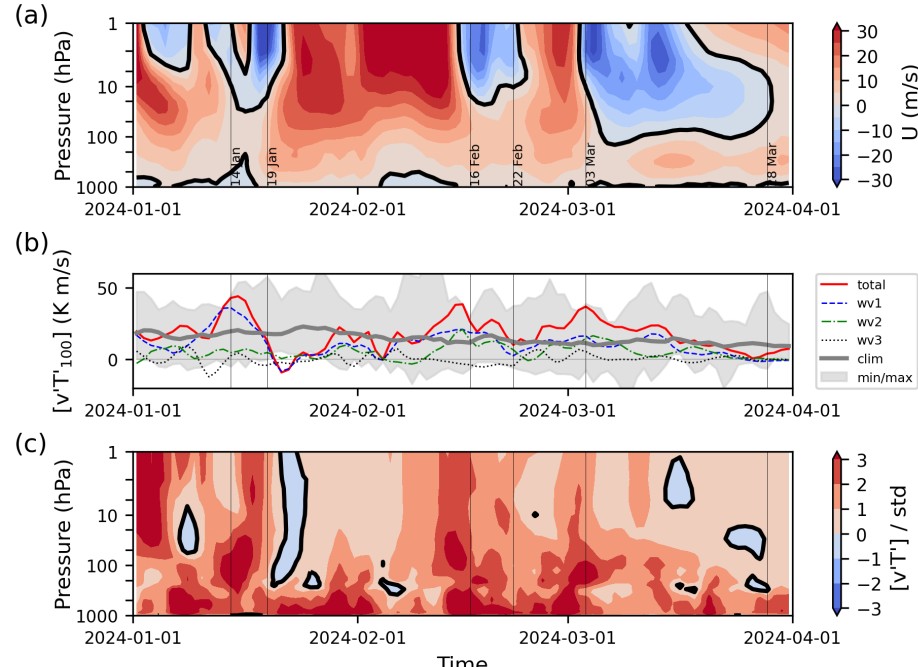

**Figure 2.** a) Time-pressure cross-section of the polar-cap averaged zonal wind, U (m/s), during January – March 2024. Thick black contours
indicate zero wind. b) 100 hPa meridional eddy heat flux area-averaged poleward of $45°$N, $[v'T'_{100}]$ (K m/s), (total, solid red) for the
same period. Zonal wavenumber 1 (wv1, dashed blue), wavenumber 2 (wv2, dashed-dotted green) and wavenumber 3 (wv3, dotted black)
components of the heat flux are calculated as described in Sect. 2. Climatological mean is plotted as thick gray curve and min/max range
is shaded based on 2000 – 2022 ERA5 data. c) Time-pressure cross-section of the total meridional eddy heat flux area-averaged poleward
of $45°$N, $[v'T']$ (K m/s), and normalized by its standard deviation, std (K m/s), at each pressure level. Thick black contours indicate zero
values. Vertical lines indicate onsets and duration of three SSWs obtained in Sect. 3.1.

Orsolini et al. (2018) investigated the differences between short- and long-lived SSWs in the ECMWF seasonal forecast
model. Their results showed that despite the westward U penetrates as deeply as in long events right at their onset, short events
decay rapidly due to the wave forcing that is less sustained than during long events. In contrast, long events continue to develop,
strengthening westward U throughout the a deepening stratospheric layer. The time-pressure cross-section of U shown for all



three SSWs in Fig. 2a, with their near-simultaneous deceleration in a deep stratospheric layer down to 20 hPa regardless of
       duration, is in agreement with (Orsolini et al., 2018).

       Figure 2b shows time series of the 100 hPa meridional eddy heat flux area-averaged poleward of 45°N. One can see three
       episodes of the increased and sustained heat flux that are collocated in time with the onsets of three SSWs. Anomalously strong
       meridional eddy heat flux at 100 hPa is known to nearly always precede weak vortex events (including SSWs), consistent with
wave–mean flow interaction theory (Polvani and Waugh, 2004; Karpechko et al., 2017). Sjoberg and Birner (2012) found that
       forcing duration has an even greater influence on SSW generation than forcing amplitude. This agrees with Fig. 2b, where
       the 100 hPa meridional eddy heat flux had increased values compared to its climatological mean for several days prior to the
       onsets. Karpechko et al. (2017) also found that SSWs are more likely to have tropospheric impact when the wave forcing at
       100 hPa is sustained for several days after the onset. Recent study by Qian et al. (2024) analyzed stratosphere-troposphere
coupling in winter 2023/24 and found that all three SSWs in winter 2023/24 were associated with positive geopotential height
       anomalies propagating down to the troposphere and the surface. This agrees well with the positive meridional eddy heat flux
       anomaly at 100 hPa remaining positive for more than five days after the onsets as seen in Fig. 2b.

       In January 2024, the zonal wavenumber 1 component of the 100 hPa meridional eddy heat flux had the largest contribution
       around the SSW onset, while the zonal wavenumber 1 and 2 components had comparable contribution around the SSW onset
in February 2024. In March 2024, two peaks in the 100 hPa meridional eddy heat flux corresponding to the two previously
       mentioned episodes of strong westward intensification of the polar-cap averaged U in the stratosphere are seen in Fig. 2a. At
       the first peak, the zonal wavenumber 1 and 2 components had nearly equal contribution, while zonal wavenumber 1 component
       dominated at the second peak. The relative contribution of the zonal wavenumber 1 and 2 components is often associated with
       the SSW type in the literature (see review by Baldwin et al., 2021, and references therein). Wavenumber 1 activity is typically
associated with displacement SSWs where the polar vortex is shifted off-the-pole, while wavenumber 2 activity is often linked
       to split SSWs where the polar vortex is divided into smaller vortices. Despite the presence of wavenumber 2 pulses in the 100
       hPa meridional eddy heat flux prior to the SSWs in February and March 2024, the recent study by Qian et al. (2024) classified
       all three SSWs in winter 2023/24 as displacement events based on the analysis of 10 hPa geopotential height time series.

       The temporal evolution of the 100 hPa meridional eddy heat flux in January 2024 largely follows the life-cycle of a short-
lived SSW as described in (Orsolini et al., 2018). Such events are characterized by a significant increase in the meridional
       eddy heat flux prior to the SSW onset followed by a quick transition to the values much below the climatological mean a few
       days after. Notably, the absolute values of the total meridional eddy heat flux became negative after the SSW onset in January
       2024 possibly indicating a reflection of PWs in the lower stratosphere. To highlight the location of a region where the total
       meridional eddy heat flux was negative, Fig. 2c shows the total meridional eddy heat flux area-averaged poleward of 45°N
and divided by its standard deviation at each pressure level as a function of pressure and time. Negative values within one
       standard deviation spread from ∼200 hPa to 1 hPa within the following days. This could dynamically accelerate the recovery
       of stratospheric circulation and may be the reason for the rapid termination of the SSW in January 2024. More details on the
       tropospheric forcing behind the negative meridional eddy heat flux is provided in Sect. 3.3.




Figure 2c also indicates several episodes of the negative meridional eddy heat flux in March 2024. In mid-March, negative
values were localized between 5 and 1 hPa around the time when the polar-cap averaged U returned to positive values in the
upper stratosphere (Fig. 2a). In the end of March, negative values of the total meridional eddy heat flux appeared in the lower
stratosphere between 70 and 20 hPa. It is evident from Fig. 2a that the westward polar-cap averaged U began to weaken around
this time and returned to the eastward regime quickly after.

### 3.3 Tropospheric forcing

Blocking high-pressure systems play a significant role in modulating the large-scale atmospheric flow, particularly through
their interactions with the tropospheric planetary waves, and control the wave activity flux into the stratosphere (Nishii et al.,
2010; Woollings et al., 2010). As mentioned in Sect. 1, such interactions have an impact on the strength of the polar vortex
based on the geographical location of BHs. Numerous blocking indices have been proposed in the literature to identify the
blocking events, and the results obtained using these indices can differ (Martius et al., 2009; Woollings et al., 2010). In this
study, we define blocking events as large ($> 0.2$ km) and persistent ($> 3$ days) positive anomalies of 200 hPa geopotential
height, and the meridional eddy heat flux $v'T'$ is decomposed following Nishii et al. (2011) as:

$$v'T' = v'_c T'_c + v'_a T'_c + v'_c T'_a + v'_a T'_a, \tag{1}$$

where the sub-indexes $c$ and $a$ denote the climatological mean and anomalies. Here, the first term corresponds to the merid-
ional eddy heat flux due to the climatological PWs. The second and third terms are linear interference terms that corresponds
to interactions between the climatological PWs and anomalies. The last term is a non-linear term.

To demonstrate the impact of the BH geographical location on the upward PW propagation variability, Fig. 3 shows the 200
hPa geopotential height anomalies together with the anomalous 100 hPa meridional eddy heat flux (i.e. the interference and
nonlinear terms in Eq. 1) in three 4-day windows around the SSW onsets in winter 2023/24. Deviations of the climatological
mean 200 hPa geopotential height from its zonal mean are also shown to highlight climatological PW ridges and troughs.

Prior to the SSW in January 2024, a large scale BH was present over the North Atlantic, Greenland and partially over the
Arctic Ocean, partially covering two PW ridges and one trough (Fig. 3a). The associated regions of positive (purple contours)
and negative (green contours) anomalous meridional eddy heat flux were observed. As seen in Fig. 2b, the combined effect
resulted in the increased 100 hPa meridional eddy heat flux area-averaged poleward of 45°N. In the following four days, the
BH began to separate into two smaller systems located over the PW troughs in the Arctic Ocean and the North Pacific causing
a reduction in the meridional eddy heat flux (Fig. 3b, and Fig. 2b). Around 5 to 8 days after the onset, the BH over the Arctic
Ocean decayed while the BH over the North Pacific sustained and migrated westward, deeper into the PW trough (Fig. 3c),
further developing negative anomalous meridional eddy heat flux. At this time, the absolute value of the 100 hPa meridional
eddy heat flux turned negative as indicated in Fig. 2b. It has already been mentioned in Sect. 3.2, that negative values of the
meridional eddy heat flux spread throughout the entire stratospheric column (see Fig. 2c). We therefore conclude that the
westward propagating BH in the North Pacific suppressed the upward PW activity and was the reason for the quick termination







**Figure 3.** Tropospheric forcing conditions around SSW in a) January, b) February, c) March 2024 for three 4-day windows around the onsets: (left) days [-3, 0], (middle) days [1, 4], (right) days [5, 8]. The 200 hPa geopotential height anomalies, $\Delta z$ (km), are shaded. Note that panels have different scales. The thin solid (dashed) black contours indicate positive (negative) deviations of the climatological mean 200 hPa geopotential height from its zonal mean (contours $\pm 0.05, 0.1, 0.15, 0.2$ km). The thick solid purple (dashed green) contours indicate positive (negative) anomalous 100 hPa meridional eddy heat flux (the interference and nonlinear terms in Eq. 1, contours $\pm 50$ K m/s). Latitude of $45^\circ$N is shown to indicate the area where the 100 hPa meridional eddy heat flux was averaged in Fig. 2.

of the SSW in January 2024. The above described timeline is in agreement with a weak polar vortex case described in (Orsolini et al., 2018).



In February 2024, tropospheric forcing conditions looked different with several BHs being present prior to the SSW onset. However, only two of them were located within the area poleward of 45°N, namely, a large BH over the PW ridge in Alaska and a small BH over the PW trough near Hudson Bay (Fig. 3d) favoring an increased meridional eddy heat flux. In the following days, a BH emerged over the lower edge of the Far East PW trough (Fig. 3e) causing increased negative anomalies in the meridional eddy heat flux and a decrease in the total 100 hPa meridional eddy heat flux (Fig. 2b). Around the SSW termination, the BH over the Far East split into 2 systems (one moved deeper into the continent and another moved eastward toward the Pacific). The BH over the Far East was lying within the PW trough and contributed to the negative anomalous heat flux (green contour in Fig. 3f). At the same time, the BH over the Pacific extended along 45°N and partially covered both PW ridge and trough. Around this time, the 100 hPa meridional eddy heat flux reduced to its climatological values as shown in Fig. 2b, allowing the polar vortex to strengthen. The presence of several high pressure systems in February 2024 could also explain the zonal wavenumber 2 signal in Fig. 2b.

In contrast to two short-lived SSWs in winter 2023/24, the forcing was more sustained around the onset of SSW in March 2024, allowing a long SSW to develop (Fig. 3g-i). Prior to the SSW onset, three high pressure systems were present poleward of 45°N: the BH in Northern Europe along the eastern edge of the PW ridge, a weaker BH over the western edge of the PW ridge in Alaska, and a small BH near the Hudson Bay PW trough. From Fig. 2b and 3g, it is clear that such geographical positioning of BHs was associated with an increased meridional eddy heat flux. Within the next four days, the BH over Northern Europe strengthened and moved towards the Greenland See, and the BH near the Hudson Bay strengthened in the PW trough. The associated decrease in the 100 hPa meridional eddy heat flux can be seen in Fig. 2b and 3h. Around 5 to 8 days after the onset, all high pressure systems seen prior to the onset weakened but stayed relatively close to their original locations, sustaining the positive values of the anomalous meridional eddy heat flux and allowing the further development of the SSW (Fig. 3i). Similar to the February SSW, the presence of several high pressure systems could explain the zonal wavenumber 2 signal in Fig. 2b. The above described sustained tropospheric forcing agrees well with what was proposed for a long SSW (Orsolini et al., 2018; Karpechko et al., 2017; Sjoberg and Birner, 2012). We therefore conclude that the persistent BH partially overlapping the PW ridge in Northern Europe enhanced the upward PW activity and was favorable for the development of a canonical long SSW in March 2024.

### 3.3.1 Contributions to the eddy heat flux near the termination of two short-lived SSWs in winter 2023/24

To further elucidate which terms dominated the anomalous meridional eddy heat flux and to decipher the roles of the background temperature and meridional wind, this section provides a deeper insight into the tropospheric forcing conditions around the termination of the two short-lived SSWs in winter 2023/24. Figures 4 and 5 provide additional details on the respective contributions of the interference and nonlinear terms in Eq. 1 to the total anomalous meridional eddy heat flux around the termination of the SSWs in January and February 2024. These spatial patterns of three terms, and of their sum, are shown in Fig. 4 for each event (left and right columns) during a 4-day time window following onset (more specifically day 5 to 8). Note that the purple and green contours in Fig. 4g-h are the same as in Fig. 3. Inspection of Fig. 4 reveals that (i) both interference terms (first and second row, respectively), can contribute to the total anomalous meridional eddy heat flux (fourth row), depending





on geographical location, and sometimes in opposite ways; (ii) The nonlinear term (third row) can also be equally important at some locations.

Figure 5 shows the time evolution of the total anomalous meridional eddy heat flux (red contours) spatially averaged over
245 mid and high latitudes, as well as of the three contributing terms (two interference and nonlinear). Again, it can be seen in these spatial averages, that the interference terms can have opposite, i.e., negative or positive, contributions and that the non-linear term can dominate at times.

In January 2024, the $v_a' T_c'$ interference term appears to be responsible for the large negative anomaly of the meridional eddy heat flux (Fig. 4a and 5 blue curve) that dominates the zonal mean. The anomalous equatorward advection of the background
(climatological) warm air over the Alaska and North Pacific, combined with the poleward advection of the background cold air over the western Eurasia are mainly responsible for the large negative anomaly of the meridional eddy heat flux (e.g., blue curve in Fig. 5). Note that the climatological temperature exhibit a clear wave-1 pattern.

There is less clarity regarding the SSW in February 2024. Fig. 5 indicates that the spatially-averaged magnitudes of the interference and nonlinear terms in Eq. 1 were all weak and of nearly equal importance in the termination phase. From Fig.
4b,d,f, it is clear that anomalous meridional wind had a complex pattern in agreement with the presence of multiple BHs in the troposphere as described above. On the other hand, the spatial pattern of the anomalous temperature had a similar wave-1 pattern as the climatological mean (Fig. 4b,d). Hence, the near-zero values of the area-averaged anomalous meridional eddy heat flux in Fig. 5 resulted largely from the mutually canceling positive and negative contributions from the meridional wind fields across the North Pacific, Eurasia, the Euro-Atlantic and North America.



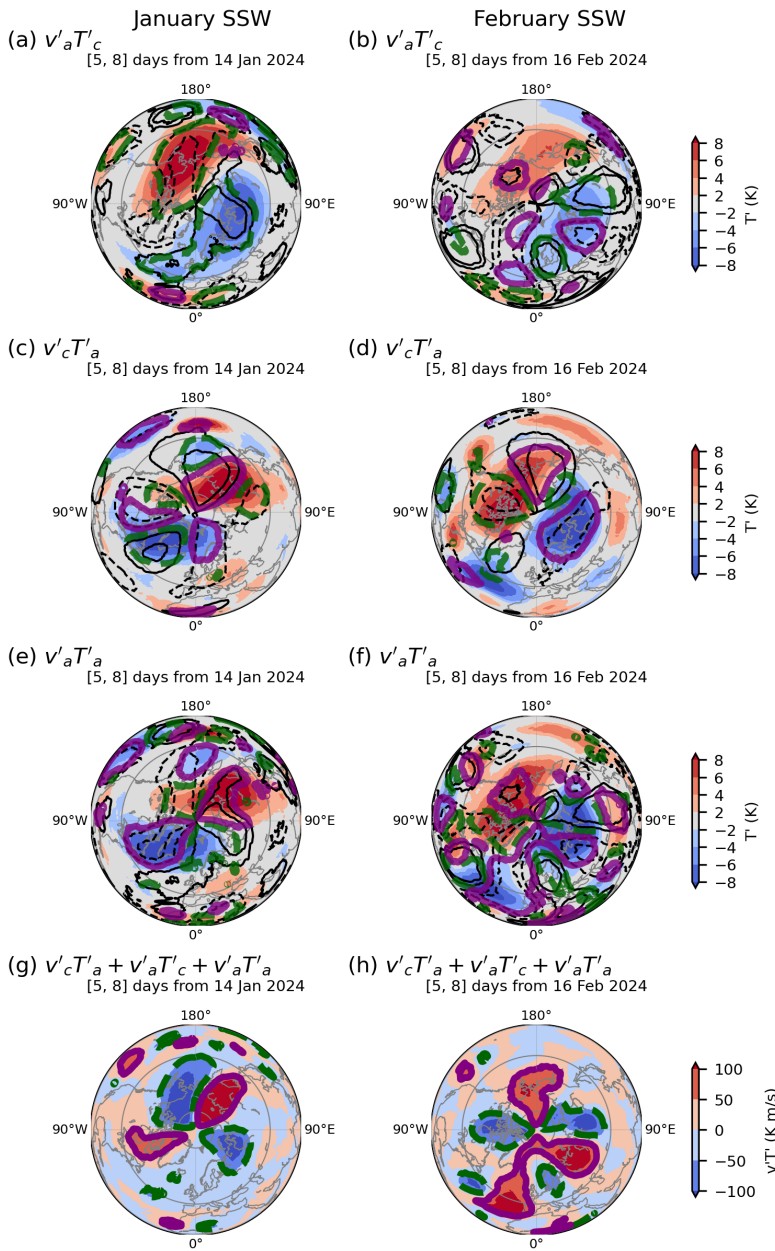

**Figure 4.** Details on the interference and nonlinear terms in Eq. 1 around the SSW termination in January and February 2024. The thick solid purple (dashed green) contours of $\pm 20$ K m/s indicate positive (negative) values of 100 hPa meridional eddy heat flux components: (a-b) $v'_a T'_c$, (c-d) $v'_c T'_a$, (e-f) $v'_a T'_a$. The eddy temperature, $T'$ (K), either climatological in (a,b) or anomalous in (c-f), is shaded and the eddy meridional wind, $v'$ (m/s), either climatological in (c-d) or anomalous in (a-b, e-f) is shown as black contours ($\pm 5, 10$ m/s). (g-h) The anomalous 100 hPa meridional eddy heat flux (sum of the interference and nonlinear terms in Eq. 1) is shaded, and positive (negative) contours of $\pm 50$ K m/s from Fig. 3 are highlighted as solid purple (dashed green) lines. Here, prime denotes the deviation from the zonal mean (or eddy component), while the sub-indexes $c$ and $a$ denote the climatological mean and anomalies.





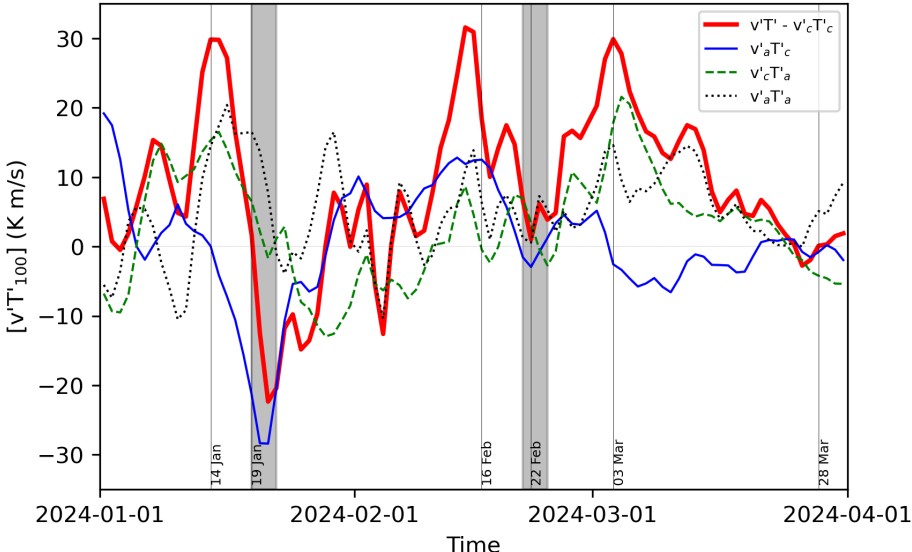

**Figure 5.** Time series of the anomalous 100 hPa meridional eddy heat flux, $v'T' - v_c'T_c'$, area-averaged poleward of 45°N (thick red) and its components $v_a'T_c'$ (solid blue), $v_c'T_a'$ (dashed green), and $v_a'T_a'$ (dotted black). The sub-indexes $c$ and $a$ denote the climatological mean and anomalies. Vertical gray stripes indicate the window of 5-8 days after the onset same as in Fig. 4.

## 4 Discussion and Conclusions

It has been previously shown in the literature that the occurrence of SSWs is influenced by external factors like Quasi-Biennial Oscillation (QBO), El Niño-Southern Oscillation (ENSO), solar cycle, and anomalous snow cover. During the easterly phase of the QBO, SSWs are shown to be more frequent Rao et al. (2019), as the low-latitude stratospheric easterly winds promote a critical line for stationary planetary waves in the subtropics, allowing more disruptions of the polar vortex (also known as the Holton-Tan relationship). ENSO also plays a key role, with the SSW likelihood being increased in both El Niño and La Niña phases (Domeisen, 2019). However, Rao et al. (2019) showed that SSWs take place more frequently in moderate El Niño winters. The interconnections between the QBO and ENSO phases can modulate the overall probability of SSW events. The 11-year solar cycle also affects the occurrence of sudden stratospheric warmings (SSWs). During solar maximum, SSWs tend to be more frequent due to the weaker and more disturbed polar vortex. The effect is opposite during the solar minimum, the polar vortex is generally stronger and more stable, reducing the chances of SSW events. Furthermore, the solar cycle influence can interact with other factors like the QBO and ENSO, further modulating the SSW likelihood. In addition, an enhanced Eurasian snow cover has been reported as a factor influencing the polar vortex variability (e.g., Cohen et al., 2007; Garfinkel et al., 2010; Henderson et al., 2018; Lü et al., 2020). Although the exact mechanism is not yet fully understood, the latest understanding of snow-stratosphere coupling is that snow anomalies, through their cooling effect, may contribute to regional modifications in near-surface temperature gradients, hence land-sea thermal contrasts conducive to the generation of PWs.



In winter 2023/24, the QBO was in its easterly phase, while the ENSO was in the El Nino phase (see Qian et al., 2024, for details). At the same time, solar cycle 25 is approaching its maximum, predicted for July 2025 according to the NOAA Space Weather Prediction Center. In addition, Vorobeva and Orsolini (2024) showed that the Northern Hemisphere snow cover exceeded its climatological mean by approximately one standard deviation in the second half of January 2024. Each of these external factors has previously been shown to increase the likelihood of SSWs, collectively creating favorable conditions for their occurrence.

A recent study by Qian et al. (2024), also based on daily ECMWF ERA5 reanalysis data, analyzed the stratosphere-troposphere coupling in winter 2023/24 including the three SSWs. As mentioned earlier, this is an extreme occurrence (Ineson et al., 2024). However, the SSW onsets were defined based on the zonal-mean U at 10 hPa and 60°N and the associated meridional temperature gradient reversal. Due to the zonal-mean U remaining positive at 60°N, the SSW in February 2024 was classified as a minor event in (Qian et al., 2024). Complementing the work of Qian et al. (2024), we conduct a detailed investigation into the mechanisms governing the duration and termination of SSWs during boreal winter 2023/24 as it remained unclear why some SSWs develop into long-lived events while others terminate rapidly.

Using the ERA5 reanalysis data and the SSW definition based on the polar-cap averaged U, we identify three major SSW events in winter 2023/24. For all three SSWs, our analysis demonstrates that blocking highs developed over the PW ridges played a crucial role in enhancing PW activity prior to onset. The first SSW occurred in January 2024, lasting from 14th to 19th. Its rapid termination and the subsequent recovery of the polar vortex are linked to a developing high-pressure system over the PW trough in the western North Pacific. The second SSW took place in February 2024, from 16th to 22nd. We find that its abrupt termination was influenced (though not exclusively driven) by a westward-propagating BH that migrated over the PW trough into the Far East. The third SSW occurred in March 2024, lasting from the 3rd to the 28th, making it a long-duration canonical SSW event. Analysis of the tropospheric forcing conditions reveals that a persistent, large-scale high-pressure system over the PW ridge in Northern Europe sustained enhanced upward PW activity, favoring the development of an extended SSW. This study highlights that subtle synoptic developments in distant oceanic basins condition the specific evolution of SSWs. In particular, synoptic developments over the Far East and the North Pacific, play an important role in the termination of short events.



*Data availability.* Hourly ERA5 data on pressure levels are available from the Climate Data Store website (Hersbach et al., 2023)

*Author contributions.* EV and YOR equally contributed to the conception of this study. EV analyzed the data and drew all figures.

*Competing interests.* The authors declare that they have no conflict of interest.

*Acknowledgements.* Authors thank the ECMWF for providing open access data for this study.



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
