# Peer review of "The impact of tropospheric blockings on duration of the sudden stratospheric warmings in boreal winter 2023/24"

_EGUsphere, 2025_

## Author Comment (AC1)

**Reply to Editor comment**

I wanted to add a minor comment regarding the use of the polar cap averaged zonal winds to detect SSWs. I think the authors provide justification for using a different definition than what is standard/most common. However, using this other definition will change how rare the occurrence of 3 SSWs actually is. As a note, the Ineson et al. 2024 paper that is cited also uses the Charlton & Polvani 2007 definition, so those statistics are not comparable to the analysis done here. Also note that Butler et al. (2015) Fig 2 shows a large increase (about 25%) in detected SSW events when using the polar-cap averaged definition. Therefore, please be careful when making statements such as "three major SSWs are identified - an extremely rare occurrence in a single winter." If you change your definition, the statistics change too, and 3 SSWs is likely not so rare when using this definition.

Dear Editor, thank you for your comment. We agree, the occurrence rate of SSWs depends on the definition used. We, therefore, made clarifications in the text when mentioning the number of SSWs in winter 2023/24.

---

## Author Comment (AC2)

**Reply to Reviewer 1 comments**

Authors thank the anonymous reviewer for taking their time to read the manuscript and for providing constructive comments. We believe the suggestions provided have helped us to improve the clarity and quality of the manuscript. Below, we address each comment in detail and describe the changes made in the revised version. Our reply is provided in blue italic font.

General statement

This research utilizes the ERA5 reanalysis data and defines the Sudden Stratospheric Warming (SSW) based on the polar-region-averaged wind field. In-depth analyses are conducted on the three major SSW events identified. The authors correctly note that the first two SSW events have relatively short durations, in contrast to the last SSW event, which persists for a substantially longer period. Additionally, they draw inferences about the relationship between the three SSW events and the tropospheric blocking pattern. The overall writing of the manuscript is commendably fluent. However, before recommending this manuscript for publication, I have several concerns that require attention.

*We thank the reviewer for the positive feedback.*

Major issues

1. The authors ascribe the rapid termination of the first SSW event to the westward-propagating Blocking High (BH) in the North Pacific (Lines 205-207). In reality, as evident from Figures 3b and 3c, the decline in V'T' is not solely observed in the North Pacific region; rather, the changes over the Atlantic Ocean are quite conspicuous. The authors should consider the causes of this weakening in a more comprehensive fashion and quantitatively present the contribution ratios of each BH.
   *We had meant that the largest contribution to the negative heat flux (thick green contours) -in a zonal mean sense- arises from the western flank of the Pacific BH, where advection is equatorward (see Fig 4g). There is also a negative contribution from Central Eurasia, but the contribution from the North Atlantic is rather positive (purple contours in Fig 4g). This discussion was clarified along these lines.*

2. Regarding the third SSW event, the authors indicate that the circulation after the outbreak is favorable for the upward propagation of planetary waves. Nevertheless, this conclusion is merely based on the average results for the 5–8 days following the outbreak and cannot represent the situation that may endure for up to one month.
   *This is correct and has now been clarified:" Note that, during and after the reversal in March, descending easterlies would hinder vertical propagation of PWs high into the stratosphere, and PWs would be evanescent in the lower stratosphere (Fig2c)".*

Minor issues

1. As can be seen from Figures 2 and 5, the intensities of planetary waves at 100 hPa one month after the occurrences of the second and third SSW events show minimal differences. Yet, one is a short-lived SSW event while the other is a long-lived one. What

accounts for this?

*This is indeed a key question of great importance for long-range forecasting. We are not sure if it was meant "one month after" here, but the persistence of PW forcing around and briefly after the onset is a key factor determining the evolution into a long or short event (Orsolini et al., 2018, cited). This longer persistence around the onset of the March event is clearly seen on Fig 2b. It is intimately linked to the synoptic evolution of BHs, as we argued.*

2.  In Figure 5, it is observable that $V_a'T_c'$ is negative during the weakening phase of each SSW event. The authors need to elucidate what factors give rise to this situation.
    *This interference term is dominated by anomalous meridional advection of the wave-1 background temperature. In fact, this term turns negative already around onset, in the JAN and MAR events. It is difficult to draw systematic conclusions: In the decay phase, it dominates in the case of the JAN event, while the other interference term or the nonlinear term dominate for the MAR event, in late March.*

3.  The rapid weakening of the upward propagation of planetary waves is a necessary condition for the short duration of the SSW. I opine that quantitatively characterizing this feature could be an important research avenue.
    *We argued that the rapid weakening or, on the opposite, the persistence of the wave forcing is intimately linked to the synoptic evolution of BHs over the different oceanic basins. Improving our understanding of these processes and their predictability would indeed be a valuable research avenue.*

---

## Author Comment (AC3)

**Reply to Reviewer 2 comments**

Authors thank the anonymous reviewer for taking their time to read the manuscript and for providing constructive comments. We believe the suggestions provided have helped us to improve the clarity and quality of the manuscript. Below, we address each comment in detail and describe the changes made in the revised version. Our reply is provided in blue italic font.

General statement

Using the reanalysis, this study analyzed the successive SSWs in the 2023/24 winter, which has been reported in a series of recent studies (Rao et al. 2025AR; Lee et al. 2025 Weather; Lu Qian et al. 2024). Especially, the study analyzed the linear interference of climatological waves and synoptic waves in the eddy heat flux. The blocking highs are emphasized. This paper is well written and I only have several minor questions.

*We thank the reviewer for the positive feedback.*

Specific comments

1. The decomposition of the eddy heat flux into different terms is necessary to better understand the wave driving. However, the anomalous eddy heat flux might be problematic. See the derivation below:

   (v'T')c = (v'cT'c+v'aT'c+v'cT'a+v'aT'a)c = v'cT'c+(v'aT'a)c

   (v'T')a = v'T' - (v'T')c = v'aT'c+v'cT'a+ [v'aT'a - (v'aT'a)c]

   This paper did not consider the contribution of (v'aT'a)c for the climatological eddy heat flux, and place this term in the total nonlinear term, which might lead to wrong conclusions.

   *Thank you for pointing out this oversight. We agree with the above formulas, following earlier work by Nishii et al. (2009). These two formulas and the reference to Nishii et al. (2009) have now been included for clarity. The climatology of the non-linear term indeed contributes to the heat flux climatology. As you noticed, the formula (1) was correct. We have recalculated the climatological flux and the anomalous heat flux nonlinear third term, to include that contribution to the climatology. As in Nishii et al. (2009), it is a small positive contribution, and this correction does not change our conclusions.*

   *This correction reflected in minute changes in the purple and green contours in Figure 3 (i.e., in the total anomalous heat flux) and in the third and fourth rows of Figure 4 ( the non-linear term and the sum of all 3 terms).*

   *And it also changed the scaling in the decomposition of the anomalous heat flux in Figure 5 (dotted black line as nonlinear term and thick red curve, which is the sum of the three terms; see Fig 5_R below which includes the climatology of the nonlinear term), but not the overall conclusions.*

[Figure]

*Figure 5_R : as original Figure 5, but with the climatology of the nonlinear term added (black triangles)*

*Figures 3-5 have been updated to show the anomalous heat flux as suggested by the reviewer. The captions and labelling in Figures 4 and 5, and the inset in Figure 5 have been corrected.*

2. L22-23: This sentence is partially true for a few SSWs that the easterlies begin to appear in the upper stratosphere. I am not sure if this statement is true for other SSWs, since the wave forcing for SSWs is primarily from the troposphere and lower stratosphere. Please insert references.

   *Reversals to easterlies initiate in the upper stratosphere or lower mesosphere, where PWs break; see Orsolini et al. (2018 their Fig. 2, cited) or Limpasuvan et al. (2016; their Fig. 2, cited) for examples using the ECWMF forecast model or the high-top WACCM model, respectively.*

3. L26-27: The SSW occurs 6-7 times every decade. Please update the SSW frequency using the modern reanalysis. Further, models also produce a frequency 6-7 every 10 years. See Rao and Garfinkel 2021 (ERL, doi: 10.1088/1748-9326/abd4fe). Baldwin et al. 2021.
   *We have changed the text to reflect this decadal occurrence.*

4. L31: Liang et al. also discussed the global impact of the SSW using model simulations (doi: 10.1007/s00382-022-06293-2).
   *This reference has now been added.*

5. L40-41: There are too many studies emphasizing the impact of the high blocking on the SSWs. Refer to Rao et al. 2018 (doi: 10.1029/2018JD028908) if necessary.

*There are indeed many case studies of blocking events and SSWs. This is why we used the abbreviation "e.g." We only cited articles laying general principles.*

6. L54: This SSW is reported in several recent studies that should be considered. Add Qian et al. 2024; Rao et al. 2025 (doi: 10.1016/j.atmosres.2024.107882); Lee et al. 2024 (doi: 10.1002/wea.7656).
*We have already cited Qian et al.,2024 multiple times throughout the paper. We have now added Rao (2025) and Lee et al. (2024), which we were not aware of at time of submission.*

7. L75: that clause => which ...
*Corrected.*

8. L93: Here you should discuss the possible impact of the SSW definition on the conclusion. Using the polar cap U, you find three SSWs. But if you use CP SSW definition, you might only pick up two SSWs. Rao et al. 2025 use three stratospheric disturbances to call those SSWs. It is not a big problem.
*Indeed, we alluded on L91 that one misses the mid-FEB event if one uses a fixed latitude for the zonal-mean wind such as 60N. This is further discussed on L109-115. It is also mentioned in the introduction now.*

9. L99: Please add sone discussion that similar figures have been shown in Lu Qian et al. 2024; Rao et al. 2025, Lee et al. 2024.
*This type of plots, as well as the dataset, are common when describing SSW events. We believe that citing all three studies is unnecessary. We added that : "Similar latitude-time cross-sections, based on ERA5 data, were shown by Qian et al. (2024), but our U line plot (Fig 1d), shows that the choice of a threshold latitude determines whether the mid-February event is classified as a major or minor SSW".*

10. L110-111: Add a reference e.g., Rao et al. 2025.
*Done. We also cite Qian et al. and Lee et al., as they also identified mid-Feb SSW as minor.*

11. L131-132: This sentence is true. Is the SSW persistency of easterlies also discussed for other two SSWs. Since the first two and especially the second SSW provide a precondition for the major SSW in March 2024, the precondition of the last SSW should be discussed.
*The persistence of the easterlies had been quantified in the duration characteristic. We have now added a sentence on the pre-conditioning of the March event (L132) : "Nevertheless, above 10 hPa, U did not reach the anomalously strong intensity it had in early February. Hence the stratospheric polar circulation became preconditioned for the March SSW."*

12. Figure 2: 45N is still too far from the polar region. Will the conclusion change if the latitude is changed to 75N.
*It is common to englobe mid-latitudes when one computes the eddy heat flux, since planetary wave activity flux encompasses mid and high latitudes (Limpasuvan et al., 2016, cited, their Figs 4 & 5; Lee et al. (2024; cited, their Fig 2).*

13. L138: What kind of forcing is persistent and what is not? Do you have any results stating?

    *In general terms, we meant the wave forcing, encapsulated in the Eliassen-Palm flux divergence (e.g., Orsolini et al., 2018, cited, their Figs 1-2), especially in the lower stratosphere. In our case, it is also seen that the eddy heat flux is more persistent during the March event (red curve, Fig 2b) in the lower stratosphere.*

14. L140: Please state what aspects are consistent and what are not.

    *We stated that the near-simultaneous deceleration in a deep stratospheric layer down to 20 hPa regardless of the short duration, is in agreement with (Orsolini et al., 2018). Based on your comment, we have now added that the lower stratospheric wave forcing persistence is also in agreement.*

15. L168: Not sure if it is true. Might also indicate that the upward propagation of waves are weakened.

    *In this case, it is not only the anomaly of the upward wave activity flux which is negative (which may indicate weakened upward propagation) but also its total value, hence the suggestion of downward propagation after PWs encountering a reflection layer.*

16. L187: Eq. 1 is right but the anomalous eddy heat flux definition might be problematic.

    *See earlier comment and corrections.*

17. L196: Please clarify where the trough is situated.

    *The trough (in the climatological sense) is located over Northern Canada. This has now been clarified.*

18. Figure 3: The climatological waves shown in black contours are not very consistent in g-i. Please check if an error exists in the plotting script. The high over 0E is not consistent, for example. The anomalous eddy heat flux is not right as I say above.

    *The eddy component from climatology is shown in contours. Panel g-I refer to different time windows. From what we can see, general locations of the PW ridges and troughs do not significantly change over the course of 12 days (three 4-day windows per SSW) to cause a concern.*

19. L208: Where are those blockings?

    *This is detailed in the next sentence. Only two blockings are poleward of 45N (Hudson Bay and Alaska). We did not mention further the BHs over the Mediterranean and the Caspian seas.*

20. L211: Far East? DO you mean East Asian trough?

    *Yes, we have replaced Far East trough by East Asian trough.*

21. L222, 226: The Alaskan blocking is not persistent at all. It weakened during the second period and reformed during the third period.

    *This is correct, we referred to the two BH over Hudson Bay and Greenland Sea; the description has been improved.*

22. L236, Figure 4: Figure 4 is too noisy and boring. I am not sure if there is necessity of showing figure 4.

*We disagree. "boring" is a subjective appreciation. For the reader who is genuinely interested in understanding the "**why and how**" the PW flux intensified or waned, Fig. 4 offers a detailed explanation in terms of the leading factors (L249-251) e.g., meridional advection of cold or warm anomalies over specific regions.*

23. Figure 5: You did not remove the contribution from the climatology of nonlinear term. Please test if this will affect the conclusion.
*See earlier comment and corrections.*

24. L262: Also see Chwat et al. 2022 doi: 10.1029/2022JD037521. All the external forcings are discussed in Chwat et al. 2022 and Rao et al. 2019.
*The latter was cited in that section in relation to the QBO, but the citation has been moved to reflect the overall influence of forcings.*

25. L273: See Rao et al. 2025AR for a review on the favorable conditions for past SSWs in last decades.
*That reference has been added, with the following sentence "For the summary of the favorable conditions for five SSWs in the last decade (2014 - 2024) see Table 1 in Rao et al. (2025)."*

26. L301: If necessary, please add the funding information. Further, the dataset availability should be shown with feasible hyperlink address.
*No external funding was used.*

---

## Author Response (AR2)

Dear Editor,

We thank you for your comment. Corresponding corrections have been made around lines 27 and 100-105.

Kind regards,

Ekaterina Vorobeva and Yvan Orsolini